# Newly Designed Quinazolinone Derivatives as Novel Tyrosinase Inhibitor: Synthesis, Inhibitory Activity, and Mechanism

**DOI:** 10.3390/molecules27175558

**Published:** 2022-08-29

**Authors:** Yaru Huang, Jiefang Yang, Yunyang Chi, Chun Gong, Haikuan Yang, Fanxin Zeng, Fang Gao, Xiaoju Hua, Zongde Wang

**Affiliations:** 1East China Woody Fragrance and Flavor Engineering Research Center of National Forestry and Grassland Administration, College of Forestry, Jiangxi Agricultural University, Nanchang 330045, China; 2Jiangxi Academy of Forestry, Camphor Engineering Research Center of National Forestry and Grassland Administration, Nanchang 330032, China; 3Yongfeng County Natural Resources Bureau, Ji’an 331500, China

**Keywords:** tyrosinase, inhibitor, citral, derivatives, fluorescence quenching, molecular docking

## Abstract

We synthesized a series of quinazolinone derivates as tyrosinase inhibitors and evaluated their inhibition constants. We synthesized 2-(2,6-dimethylhepta-1,5-dien-1-yl)quinazolin-4(3*H*)-one (Q1) from the natural citral. The concentration, which led to 50% activity loss of Q1, was 103 ± 2 μM (IC_50_ = 103 ± 2 μM). Furthermore, we considered Q1 to be a mixed-type and reversible tyrosinase inhibitor, and determined the K_I_ and K_IS_ inhibition constants to be 117.07 μM and 423.63 μM, respectively. Our fluorescence experiment revealed that Q1 could interact with the substrates of tyrosine and L-DOPA in addition to tyrosinase. Molecular docking studies showed that the binding of Q1 to tyrosinase was driven by hydrogen bonding and hydrophobicity. Briefly, the current study confirmed a new tyrosinase inhibitor, which is expected to be developed into a novel pigmentation drug.

## 1. Introduction

Tyrosinase is a multifunctional enzyme containing binuclear copper [1]. It can catalyze the transformations of *o*-diphenols or monophenols toward the suitable quinones, which then afford melanin [2]. Thus, it plays an important role in the biosynthesis of melanin. Melanin is a significant pigment distributed in an animal’s hair and skin [3]. However, an overdose of melanin may cause skin cancer [4,5,6]. Moreover, a serious agricultural problem caused by tyrosinase is the browning enzymatic reaction, resulting in the color and quality degradation of vegetables and fruits in the processes of storage and postharvest [7,8]. Thus, an inhibition of melanin biosynthesis can be used not only to prevent skin diseases but also to develop skin-lightening agents for aesthetic concerns [9,10,11]. The tyrosinase inhibitors thus meet cosmetic demands, and also provide a foundation for the development of hyperpigmentation drugs. Thus, they are of considerable interest in agriculture, food, medical, and cosmetic industries.

Over the last few decades, many techniques have been used to search for antityrosinase potentials, including molecular dynamics simulation, conformational analysis, and biosilico identification. However, only a few have application potential because of cytotoxicity and carcinogenicity [4,7]. Hence, the demand for tyrosinase inhibitors is constantly increasing, and a series of compounds were isolated or synthesized from natural origins. Their inhibitory activities on tyrosinase have also been studied by in vitro and in silico applications [12,13,14,15,16].

Citral is obtained from multiple plant species. In addition to being an important odor component in cosmetics and foods, citral also shows good biological activities in vitro, including antibacterial, antifungal, antioxidant, and anti-inflammatory effects [17,18,19]. Furthermore, citral has potential therapeutic significance as a local anesthetic and relaxant for smooth muscle due to its capability of promoting the relaxation of uterine, tracheal and aortic smooth muscle [20,21,22,23]. However, the interest for citral as an anti-tyrosinase agent has always been considerably lower and under-investigated. On the other hand, quinazolinones have exhibited inhibitory activity on several enzymes, such as glucuronidase, xanthine oxidase, thymidine phosphorylase, carbonic anhydrase II, phosphodiesterase I, and so on [24,25,26,27,28]. Recently, there have been reports of quinazolinone derivatives being used as tyrosinase inhibitors. In this respect, a series of quinazolinone derivatives have been designed, synthesized, and evaluated for their inhibitory activity against mushroom tyrosinase. The substituents at position 2 of quinazoline have a large effect on their inhibition of tyrosinase activity [29,30,31,32]. The strategy of hybridizing two or more pharmacophores into one molecule has attracted much attention from medicinal chemists due to their potential to expand biological activity, overcome drug resistance, and reduce toxic effects [33,34]. Therefore, hybridizing quinazolinones with other active molecules such as citral will undoubtedly be one of the effective ways to obtain novel anti-tyrosinase drugs.

Hence, we synthesized and obtained a series of substituted quinazolinone derivatives at position 2, and estimated its inhibitory activities on tyrosinase in vitro. We considered 2-(2,6-dimethylhepta-1,5-dien-1-yl)quinazolin-4(3*H*)-one (Q1) synthesized from the natural citral as the potent inhibitor on tyrosinase. Although citral has been confirmed to exhibit anti-tyrosinase activity [35], the inhibitory mechanisms and activities of new citral derivatives on tyrosinase have never been covered. The objective of our study was to investigate and compare the inhibitory activities and mechanisms of new citral-quinazolinone derivatives on tyrosinase. Citral is an important forest resource in China. Thus, the current research results will be conducive to the development of new whitening agents from citral. Additionally, this study has a positive significance for developing the intensive processing of citral resources and promoting the utilization of forest resources.

## 2. Results and Discussion

### 2.1. Chemistry

Under the exhibited synthetic methods reported in ref. [36], natural citral or aromatic aldehydes were readily applied to afford the desired quinazolinone derivatives in moderate to excellent yields (Table 1). When citral was applied to the protocol, the desired products were obtained in moderate yields (49–61%). In the absence of a substituent on the quinazoline mother ring, the reaction efficiency was relatively better, and the yield could reach 61%. Both the electron-withdrawing (F, Cl) and the electron-donating (CH_3_) groups had a negative impact on the reaction. Aromatic aldehydes could afford the desired products in high yields of between 73 and 81%. The electronic effects of the aromatic aldehydes had little effect on the formation of products. This method could be employed on a gram scale without any loss of efficiency.

### 2.2. Enzyme Inhibition Studies

Firstly, the effect of the Q1compound against the tyrosinase activity was evaluated. In Figure 1, we show that the oxidation of L-DOPA catalyzed by tyrosinase decreased with the increasing concentration of Q1. The concentration leading to a 50% activity loss (IC_50_) was 103 ± 2 μM, which was superior to the standard inhibitor arbutin (IC_50_ = 180 μM) [37]. The inhibitory activity of Q1 could not be compared with that of flavonoids [38,39,40,41]; however, it surpassed that of acrylamide and basilinearomatic [42,43,44]. In our research, we synthesized several structurally similar quinazolinone derivatives to evaluate the inhibitory activity of tyrosinase—the results are listed in Table 1. Compounds synthesized from citral exhibited latent inhibitory activity (Q2, Q3, and Q4). For example, substrates Q2 and Q3 with the electron-withdrawing group on the quinazoline mother ring were selected for the inhibitory tyrosinase activity assays, and the IC_50_ values were larger compared with that of Q1. Note that the IC_50_ value of substrate Q4, which contained the electron-donating group, was even bigger than that of Q2 and Q3. These experimental results showed that the substitution groups on the quinazoline mother ring had a great effect on the inhibition of the compound’s tyrosinase activity. Additionally, a mother ring with no substituent may be the best option. However, when the position 2 on the quinazoline mother ring was substituted by aryl groups, it did not exhibit inhibitory activity against tyrosinase (Q5, Q6, and Q7). Overall, Q1 inhibited the best activity; thus, the inhibitory activity against Q1 tyrosinase was further evaluated.

### 2.3. Kinetic Studies

To ascertain the inhibition type and mechanism of Q1, we studied the relationship between the enzyme concentration and the remaining enzyme activity with various concentrations of Q1 (0, 15, 60, and 120 μM). Additionally, the plots of the enzyme activity versus the concentrations of enzyme in various amounts of Q1 are exhibited in Figure 2. The results revealed that all the lines intersected with the origin and demonstrated a good linear relationship. Furthermore, the slopes of the lines decreased with increasing concentrations of Q1, indicating that Q1 did not reduce the amount of mushroom tyrosinase; instead, it resulted in a decrease in activity in the enzyme catalyzing the L-DOPA oxidation. These results clearly revealed that Q1 could inhibit mushroom tyrosinase in a reversible way.

Different concentrations of the compound Q1 (0, 15, 60, and 120 μM) were applied to inhibit tyrosinase and obtain the kinetic behavior of tyrosinase catalysis. Lineweaver–Burk reciprocal plots were used to investigate the inhibition modes of Q1 on mushroom tyrosinase. As shown in Figure 3, a series of lines with various intercepts and slopes met in the second quadrant, revealing Q1 inhibited enzyme activity adopting a mixed inhibition type. Thus, Q1 could bind not only to the free enzymes but also to the enzyme–substrate complexes. In addition, we obtained the inhibitor constant (K_I_) through the concentrations of the compound versus the plots of the slope, and the value of K_I_ was determined to be 117.07 μM. Additionally, the value of K_IS_ was measured to be 423.63 μM from the concentrations of the compound versus the vertical intercept. The value of K_IS_ was higher than K_I_, revealing that Q1 has a stronger affinity for the free enzyme than the enzyme–substrate complex.

### 2.4. Fluorescence Analysis

#### 2.4.1. Fluorescence Emission Spectra of Mushroom Tyrosinase in the Presence of Q1 with Various Concentrations

The intrinsic interaction mechanisms of Q1 on the tyrosinase were investigated by the fluorescence quenching analysis. The fluorescence emission spectrum was recorded as ranging from 310 to 550 nm at a 290 nm excitation wavelength. As shown in Figure 4, tyrosinase had a strong fluorescence peak at 333 nm because of fluorophore tryptophan residue, while Q1 displayed an emission peak at 411 nm. However, the fluorescence intensity of tyrosinase reduced gradually as the concentration of the citral derivative increased. When Q1 was added to 36 μM, the relative fluorescence intensity of tyrosinase decreased to 39.45%. Interestingly, raising the concentrations of Q1 resulted in a red shift, and the fluorescence emission intensity of Q1 disappeared in the spectra. These results showed that, as a good fluorescence quencher, Q1 could combine with the tyrosinase, changing the enzyme conformation and decreasing the enzyme activity.

#### 2.4.2. Fluorescence Emission Spectra of Tyrosine in the Presence of Q1 with Various Concentrations

We also investigated the fluorescence quenching Q1 experiment regarding tyrosine activity, and recorded the fluorescence intensity of tyrosine in the range of 550–700 nm. As shown in Figure 5, the fluorescence emission wavelength of compound Q1 (583 nm) was very close to that of tyrosine (586 nm). With the concentration of Q1 increasing from 0 to 36 μM, the fluorescence intensity decreased from 447.6 to 216.5 (by 51.63%). Thus, the compound was also a good binding agent for tyrosine in a dose-dependent manner.

#### 2.4.3. Fluorescence Emission Spectra of L-DOPA in the Presence of Q1 with Various Concentrations

Finally, the fluorescence emission spectra of L-DOPA were tested in the solutions of Q1 with various concentrations. As exhibited in Figure 6, L-DOPA had a peak at 314 nm; in contrast, the fluorescence emission of Q1 displayed a much weaker intensity. Similar to previous test results, the fluorescence intensity decreased from 1619 to 879.9 (by 45.65%), with the concentrations of Q1 changing from 0 to 36 μM. Similarly, the fluorescence emission intensity of Q1 disappeared when it was added to the solution. The results inferred that Q1 was also a good L-DOPA binding agent.

### 2.5. Molecular Docking

To obtain better a comprehension of the mutual effects between citral derivatives and tyrosinase, molecular docking was performed using the ActoDock tool (1.5.6). Of all conformations, the one with the lowest free energy was considered the most active optimal conformation. As shown in Figure 7, hydrogen bonds were generated between Q1 and tyrosinase residues: hydrogen bonds existed between O and N atoms on the quinazolinone ring and tyrosinase residues on SER146, with bond lengths of 3.38 Å and 2.27 Å, respectively. There was a hydrogen bond between position 1 N on the quinazolinone ring and tyrosinase residue GLY199, with a bond length of 3.38 Å. There were hydrophobic bonds between two methyl groups at the end of the citral chain and the tyrosinase residue ALA110, with bond lengths of 4.41 Å and 3.18 Å, respectively. There was a hydrophobic bond between the methyl group in the citral chain and the tyrosinase residue ALA202, and the bond length was 3.43 Å. These results vividly indicated that the binding of Q1 to tyrosinase was driven by hydrogen bonding and hydrophobicity.

## 3. Experimental Section

### 3.1. Synthesis of Quinazolinone Derivatives

We obtained natural citral from Jiangxi Sipaisi Spice Chemical Co., Ltd. We synthesized quinazolinone derivatives from the condensation of *o*-amino-benzamide and natural citral or benzaldehydes using the classical quinazolinone synthesis method. We added the *o*-amino-benzamides (5.0 mmol), citral or benzaldehydes (6.0 mmol), and 10 mL DMSO into a 50 mL round-bottom flask. We carried out the reaction overnight at 100 °C, and monitored the TLC until the reaction was completed (about 20 h). After the reaction was completed, we cooled the reaction liquid to room temperature. Then, we added 10 mL H_2_O to the reaction liquid, which we extracted with ethyl acetate. We dried the organic layer with anhydrous sodium sulfate, and removed the solvent by decompression. We used volume chromatography (eluent: petroleum ether: ethyl acetate = 5:1) to afford the desired compound. We recorded NMR spectra on Bruker AV400 spectrometer (Bruker, Billerica, MA, USA), and HRMS on Waters Xevo G_2_-S QTof/Tof ACQUITY UPLC H-Class spectrometry (Waters, Milford, MA, USA). The structural characterization data of the products are as follows:

*2-(2,6-dimethylhepta-1,5-dien-1-yl)quinazolin-4(3H)-one* (Q1). white solid; yield 61%; mp = 192–193 °C; HRMS (ESI) *m*/*z* [M + H]^+^ calcd for C_17_H_21_N_2_O^+^ 269.1654, found 269.1648; ^1^H NMR (400 MHz, CDCl_3_) δ: 11.56 (s, 1H), 8.33–8.24 (m, 1H), 7.79–7.68 (m, 2H), 7.49–7.39 (m, 1H), 6.14 (s, 1H), 5.19 (s, 1H), 2.38–2.22 (m, 7H), 1.71 (s, 3H), 1.67 (s, 3H); ^13^C NMR (100 MHz, CDCl_3_) δ: 164.1, 164.0, 154.6, 151.5, 149.7, 134.6, 132.5, 127.6, 126.2, 123.3, 120.4, 117.2, 41.4, 26.4, 25.7, 19.4, 17.8.

*2-(2,6-dimethylhepta-1,5-dien-1-yl)-5-fluoroquinazolin-4(3H)-one* (Q2). white solid; yield 51%; mp = 189–191 °C; HRMS (ESI) *m*/*z* [M + H]^+^ calcd for C_17_H_2__0_FN_2_O^+^ 287.1560, found 287.1567; ^1^H NMR (400 MHz, CDCl_3_) δ 11.43 (s, 1H), 7.65 (m, 1H), 7.50 (d, *J* = 8.4 Hz, 1H), 7.08–7.01 (m, 1H), 6.10 (s, 1H), 5.16–5.14 (m, 1H), 2.33 (d, *J* = 1.2 Hz, 3H), 2.29 (s, 4H), 1.70 (s, 3H), 1.65 (s, 3H); ^13^C NMR (100 MHz, CDCl_3_) 162.8, 161.7, 160.2, 156.1, 152.6, 152.0, 134.9, 134.8, 132.7, 123.6, 123.5, 123.3, 116.8, 112.7, 112.5, 110.2, 110.2, 41.6, 26.4, 25.8, 19.7, 17.8.

*6-chloro-2-(2,6-dimethylhepta-1,5-dien-1-yl)quinazolin-4(3H)-one* (Q3). white solid; yield 49%; mp = 190–192 °C; HRMS (ESI) *m*/*z* [M + H]^+^ calcd for C_17_H_2_ClN_2_O^+^ 303.1264, found 303.1268; ^1^H NMR (400 MHz, CDCl_3_) δ 11.68 (s, 1H), 8.22 (d, *J* = 1.2 Hz, 1H), 7.67–7.64 (m, 2H), 6.11 (s, 1H), 5.19 (s, 1H), 2.34–2.30 (m, 7H), 1.71 (s, 3H), 1.68 (s, 3H); ^13^C NMR (100 MHz, CDCl_3_) 163.3, 155.9, 151.8, 148.4, 135.1, 132.8, 132.0, 129.4, 125.6, 123.2, 121.4, 117.0, 41.7, 26.5, 25.8, 19.7, 18.0.

*2-(2,6-dimethylhepta-1,5-dien-1-yl)-7-methylquinazolin-4(3H)-one* (Q4). white solid; yield 54%; mp = 188–192 °C; HRMS (ESI) *m*/*z* [M + H]^+^ calcd for C_1__8_H_2__3_N_2_O^+^ 283.1810, found 283.1813; ^1^H NMR (400 MHz, CDCl_3_) δ 11.07 (s, 1H), 8.15 (d, *J* = 8.0 Hz, 1H), 7.51 (s, 1H), 6.08 (s, 1H), 5.17 (s, 1H), 2.50 (s, 3H), 2.29–2.18 (m, 8H), 1.71 (s, 3H), 1.66 (s, 3H); ^13^C NMR (100 MHz, CDCl_3_) 163.8, 154.4, 151.5, 150.0, 145.7, 132.6, 128.0, 127.5, 126.1, 123.4, 118.1., 117.5, 41.5, 26.5, 25.8, 22.1, 19.5, 17.9.

*2-phenylquinazolin-4(3H)-one* (Q5). White solid; yield 73%; mp 235–237 °C; HRMS (ESI) *m*/*z* [M + H]^+^ calcd for C_1__4_H_11_N_2_O^+^ 233.0871, found 233.0873; ^1^H NMR (400 MHz, DMSO) δ 12.53 (s, 1H), 8.24–8.14 (m, 3H), 7.84 (dt, *J* = 7.6, 1.6 Hz, 1H), 7.75 (d, *J* = 8.0 Hz, 1H), 7.64–7.51 (m, 4H); ^13^C NMR (100 MHz, DMSO) δ 162.7, 151.8, 148.2, 135.1, 132.2, 132.8, 128.1, 127.2, 128.9, 127.5, 126.2, 120.4.

*2-(p-tolyl)quinazolin-4(3H)-one*
(Q6). White solid; yield 81%; HRMS (ESI) *m*/*z* [M + H]^+^ calcd for C_1__5_H_13_N_2_O^+^ 237.1028, found 237.1023; ^1^H NMR (400 MHz, DMSO) δ 12.45 (s, 1H), 8.15 (dd, *J* = 8.0, 0.8 Hz, 1H), 8.12 (d, *J* = 8.0 Hz, 2H), 7.84 (t, *J* = 8.0 Hz, 1H), 7.74 (d, *J* = 8.0 Hz, 1H), 7.52 (t, *J* = 7.6 Hz, 1H), 7.35 (d, *J* = 8.0 Hz, 2H), 2.41 (s, 3H); ^13^C NMR (100 MHz, DMSO) δ 162.6, 152.6, 149.2, 141.8, 135.1, 130.3, 129.5, 128.1, 127.8, 126.7, 125.3, 121.3, 21.3.

*2-(4-methoxyphenyl)quinazolin-4(3H)-one* (Q7). White solid; yield 79%; HRMS (ESI) *m*/*z* [M + H]^+^ calcd for C_1__5_H_13_N_2_O_2_^+^ 253.0977, found 269.0973; ^1^H NMR (400 MHz, DMSO) δ 12.41 (s, 1H), 8.21 (d, *J* = 8.8 Hz, 2H), 8.14 (dd, *J* = 8.0, 0.8 Hz, 1H), 7.72 (dt, *J* = 7.6, 1.2 Hz, 1H), 7.73 (d, *J* = 8.4 Hz, 1H), 7.48 (t, *J* = 8.0 Hz, 1H), 7.11 (d, *J* = 8.8 Hz, 2H), 3.85 (s, 3H); ^13^C NMR (100 MHz, DMSO) δ 162.8, 162.4, 151.3, 148.4, 134.0, 128.9, 126.8, 126.5, 126.3, 124.3, 120.2, 114.6, 56.9.

### 3.2. Enzyme Assay

We purchased mushroom tyrosinase (with specific activity 6680 U/mg), and L-3,4-dihydroxyphenyl-alanine (L-DOPA) from Sigma (St. Louis, MO, USA). We measured enzyme assay with L-DOPA as the substrate. We determined enzyme activity at 475 nm in the spectrophotometric experiments. We dissolved the test compounds in DMSO at 1 mM, which was diluted to the required concentration (0, 15, 30, 60, 120, 180, 240, 300 μM). We measured the reaction solution (3 mL), consisting 0.75 mL sodium phosphate buffer (0.1 M, pH 6.8), 1.8 mL H_2_O, 0.3 mL L-DOPA (2 mM), 0.1 mL test compound, and 0.05 mL enzyme (0.2 mg /L), at 30 °C. We replaced test samples as control. We determined each concentration in three experiments. We used the concentration that inhibited half of the enzyme activity (IC_50_) to measure the inhibitory effect of inhibitor. We investigated kinetic parameters with increasing concentrations of L-DOPA and test samples by the Lineweaver–Burk double-reciprocal plot and Dixon plot methods. We calculated the % inhibition of tyrosinase as follows: Inhibition (%) = [1 − B/A]/100, where the A and B are the absorbances of the blank control and samples, respectively.

### 3.3. Fluorescence Analysis

We used fluorescence analysis (Varian Cary Eclipse fluorescence spectrophotometer) to perform fluorescence experiment, and set the excitation wavelength at 290 nm. We mixed inhibitor solution, substrates of tyrosine/L-DOPA or enzyme solution, and sodium phosphate buffer (pH 6.8) at the volumetric ratio of 1:1:8 (*v*/*v*/*v*).

### 3.4. Molecular Docking

We structured the initial model for docking simulations of the oxy tyrosinase from streptomyces castaneoglobisporus. We generated the 3D structures of citral derivative by Chem Bio Draw Ultra 8.0. We achieved docking calculations by using the default parameters. We selected the docked conformation with lowest free energy as the optimal binding pattern.

## 4. Conclusions

In summary, in this study, we evaluated a series of quinazolinone derivates as tyrosinase inhibitors and determined their inhibition constants. We considered 2-(2,6-dimethylhepta-1,5-dien-1-yl)quinazolin-4(3*H*)-one (Q1) synthesized from citral to be a mixed-type and reversible tyrosinase inhibitor. Furthermore, fluorescence analysis suggested that the Q1compound could react with the mushroom tyrosinase and combine with the substrates (tyrosine and L-DOPA). Therefore, Q1 could impact the melanin biosynthesis process and cut off melanogenesis by combining with substrates. Molecular docking studies revealed that the binding of Q1 to tyrosinase was driven by hydrogen bonding and hydrophobicity. Hence, Q1 is expected to be developed into a novel pigmentation drug.

## Figures and Tables

**Figure 1 molecules-27-05558-f001:**
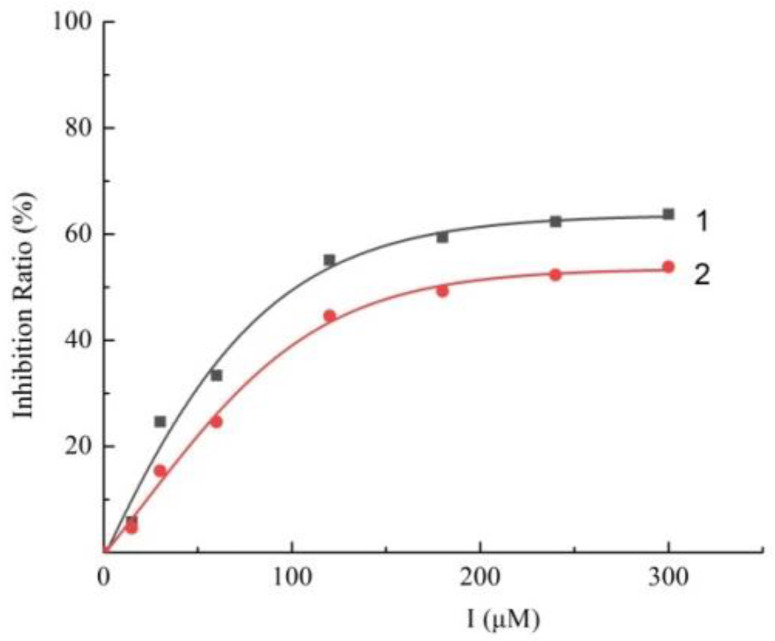
Inhibitory effect of Q1 (curve 1) and arbutin (curve 2) on mushroom tyrosinase.

**Figure 2 molecules-27-05558-f002:**
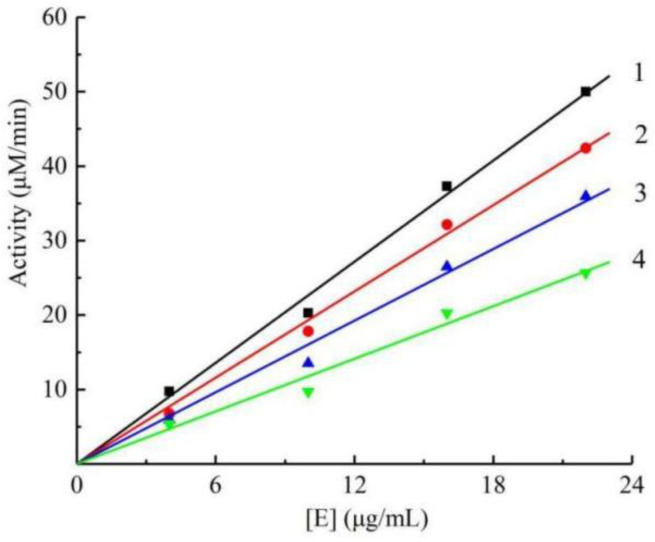
Remaining mushroom tyrosinase activity at different concentrations of Q1. Concentrations of Q1 for curves 1–4 were 0, 15, 60, 120 µM.

**Figure 3 molecules-27-05558-f003:**
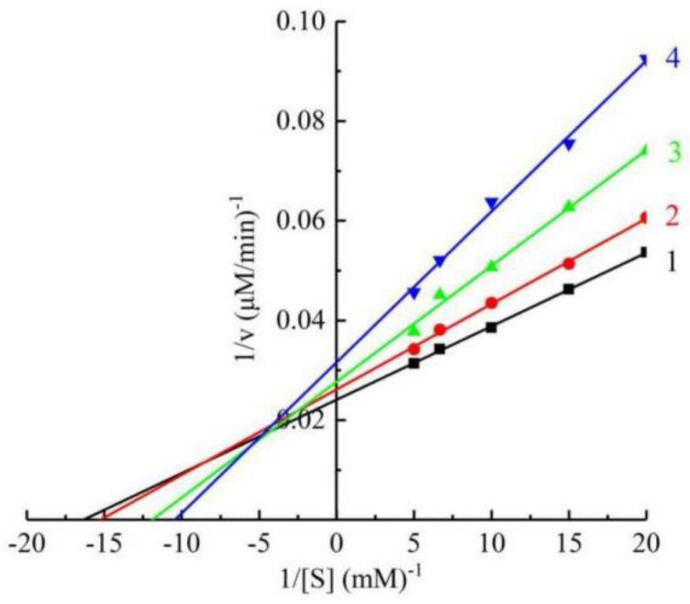
Kinetic parameters of mushroom tyrosinase inhibition by Q1. Concentrations of Q1 for curves 1–4 were 0, 15, 60, 120 µM.

**Figure 4 molecules-27-05558-f004:**
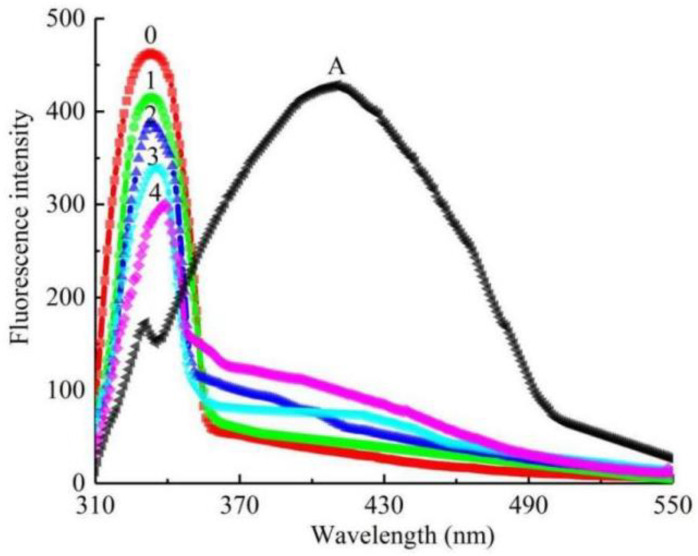
Fluorescence emission spectra of mushroom tyrosinase solution in presence of Q1 with different concentrations. Curves 0–4 representing concentrations of Q1 are 0, 9, 18, 27, 36 μM; curve A is fluorescence emission spectra of Q1 at concentration of 36 μM.

**Figure 5 molecules-27-05558-f005:**
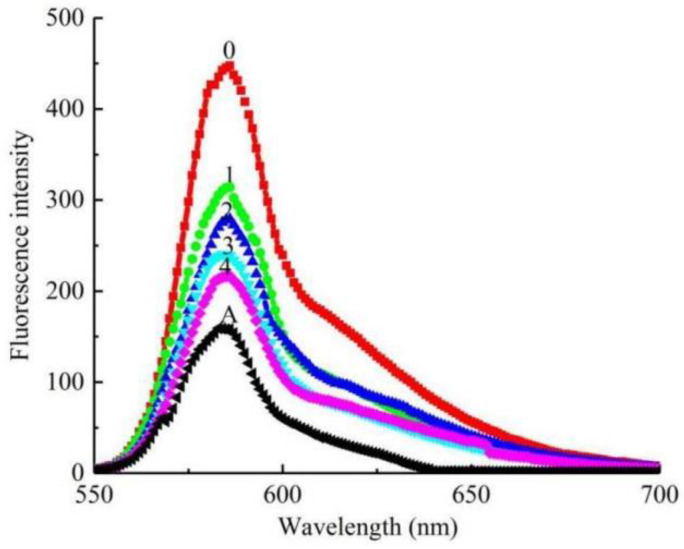
Fluorescence emission spectra of tyrosine solution in presence of Q1 with different concentrations. Concentrations of samples for curves 0–4 are 0, 9, 18, 27, 36 μM. Curve A is fluorescence emission spectra of Q1 at concentration of 36 μM.

**Figure 6 molecules-27-05558-f006:**
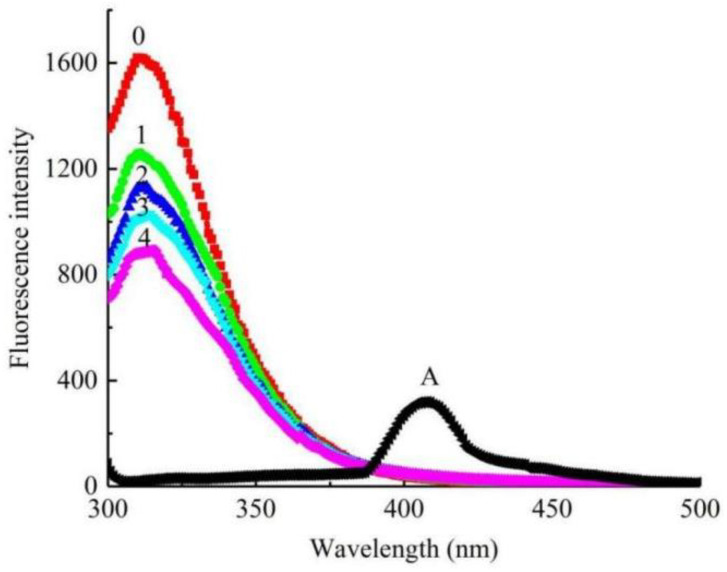
Fluorescence emission spectra of L-DOPA solution in presence of Q1 with different concentrations. Concentrations of samples for curves 0–4 are 0, 9, 18, 27, 36 μM. Curve A is fluorescence emission spectra of Q1 at concentration of 36 μM.

**Figure 7 molecules-27-05558-f007:**
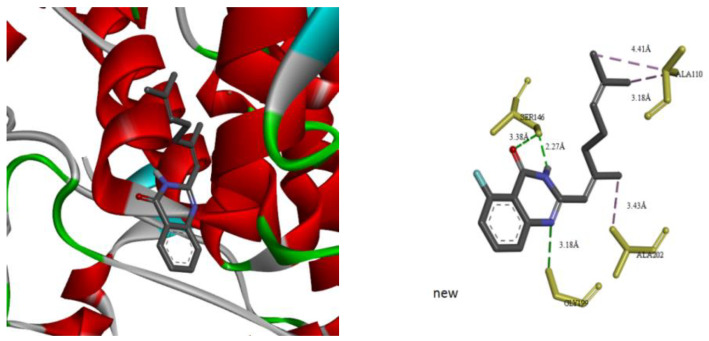
Molecular docking result of Q1 with mushroom tyrosinase.

**Table 1 molecules-27-05558-t001:** Synthesis and inhibition of quinazolinone derivatives on tyrosinase.

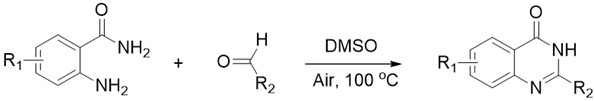
Compound	Structure	Yield *^a^* (%)	IC_50_ *^b^* (µM)
Q1	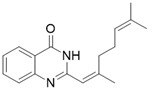	61%	103 ± 2
Q2	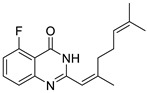	51%	105 ± 1
Q3	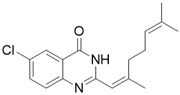	49%	168 ± 2
Q4	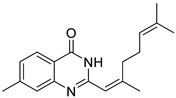	54%	253 ± 2
Q5	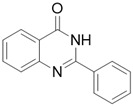	73%	NA
Q6	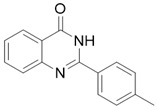	81%	NA
Q7	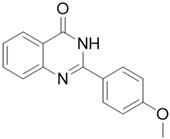	79%	NA

*^a^* Isolated yields. *^b^* Values are expressed as mean of triplicate determinations ± standard deviation.

## Data Availability

The data presented in this study are available on request from the corresponding author.

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
