# Peer review of "Newly Designed Quinazolinone Derivatives as Novel Tyrosinase Inhibitor: Synthesis, Inhibitory Activity, and Mechanism"

_molecules, 2022, doi:10.3390/molecules27175558_

Round 1
Reviewer 1 Report (New Reviewer)
The manuscript entitled “2-(2,6-dimethylhepta-1,5-dien-1-yl)quinazolin-4(3H)-one as a novel tyrosinase inhibitor: synthesis, inhibitory activity, and mechanism” represents the preparation of seven quinazolinone derivatives and to investigate their inhibitory effect on tyrosinase. The overall merit of the presented work is so low and, as a result, I do not recommend publication of the manuscript in its current format as it will be of minor interest to the readers of Molecules.
According to the authors, the objective of the study presented is to investigate and compare the inhibitory activity and mechanisms of the synthesized citral derivatives on the tyrosinase, however, not all the presented structures have a citral moiety as part of their structure. The authors have not described the basis of their selection to the structures presented. To me, the selection does not reflect a systematic design of the structures screened and selection was completely random. The library of compounds screened for the intended activity is also too narrow for a conclusion to be drawn. A control substance was not included in the study to benchmark the effect of Q1 (the structure that gave the highest effect). Chemistry-wise, the reaction used in not novel, its mechanism is well known, and the library of compounds prepared is not broad (only seven compounds). Some references are also missing.
Author Response
Please see the attachment.

Reviewer 2 Report (New Reviewer)
The manuscript sent to Molecules for evaluation (prior to publication) has as main subject the use of a title compound as a novel tyrosinase inhibitor, and the manuscript presents the organic synthesis, the inhibitory activity and the mechanism of action. The work is well presented and ends with 37 references. However, before publication, there are some issues to be solved:
-Results and discussion starts directly with the compounds characterization, at least one paragraph dealing with the synthesis is necessary;
-2.1 subchapter has to be moved to 3 chapter, the Experimental one, and therefore all the numbering will be altereted and needs to be in the correct order;
-Chart 1 has to be moved instead of subchapter 2.1 and make it bigger in size;
-text can be improved, making it more attractive, as a feeling of routine work emerges.
The manuscript may present importance for some scientists working in the area of enzymology and also for some preparative organic chemists. In conclusion, the work can be published just after these small corrections are made (see up the comments).
Author Response
Please see the attachment.

Reviewer 3 Report (New Reviewer)
In this study, the authors evaluated a series of quinazolinone derivates as tyrosinase inhibitors and determined their inhibition constants. Fluorescence experiment revealed that 2-(2,6-dimethylhepta-1,5-dien-1-yl)quinazolin-4(3H)-one (Q1) could interact with the substrates of tyrosine and L-DOPA in addition to tyrosinase. Molecular docking studies showed that the binding of Q1 to tyrosinase was driven by hydrogen bonding and hydrophobicity. This study confirmed Q1 to be a new tyrosinase inhibitor.
Overall, it is a good manuscript, but some revisions are needed before publication.
I have the following suggestions.
1) Title of the article should be corrected. This manuscript describes the synthesis and evaluation of series of quinazolinone derivatives not just Q1. The title of the manuscript should reflect that.
2) These compounds are hybrid between quinazolinone and citral molecules. There should be a brief discussion on the use of hybrid molecules as potential drugs in the introduction section.
3) The end of the introduction section is hastily written. Needs to elaborate a bit more.
4) The synthetic section description can be improved.
5) The authors provided strong spectral analysis data, as well as enzyme kinetics, which strongly supports the conclusion. The only drawback noted is that the authors compared IC50 of Q1 with standard inhibitor arbutin, but it is not clear if they had included arbutin in the same experiment. My strong recommendation is that the authors should include arbutin data in figure1, for a better understanding".
6) Minor: English can be improved
Round 2
Reviewer 1 Report (New Reviewer)
I have reviewed the comments provided by the authors, however, I am still sticking to my feedback provided and still would not recommend publication in the current format.
Author Response
Dear reviewer
Thank you for your useful comments and suggestions on our manuscript. Quinazolinones have exhibited inhibitory activity on several enzymes. Hence, we synthesized and obtained a series of substituted quinazolinone derivatives at position 2, and estimated the inhibitory activities on tyrosinase in vitro. Among them, Q1 synthesized from the natural citral was considered as the potent inhibitor on tyrosinase. Thus, the inhibitory activity against tyrosinase of the Q1 was further evaluated. On the other hand, the interest for citral as the anti-tyrosinase agents has always been considerably lower and under-investigated. Thus, the current research results will be conductive to the development of new whitening agents from citral.
We hope that our manuscript meets the high standards of the journal. We are looking forward to receiving a positive response from you regarding our manuscript.
Best wishes.
This manuscript is a resubmission of an earlier submission. The following is a list of the peer review reports and author responses from that submission.
Round 1
Reviewer 1 Report
This is a poor paper investigating the inhibition of mushroom tyrosinase. The authors have synthesised one compound which is novel. No details at all in the results and discussion about how and why the compound was synthesised. The abstract suggests several compounds but results only document one. The manuscript contains many typographical and grammatical errors. There is little justification for the study.
The dose-response curve is inappropriate. The authors need to plot log10 molar concentrations of inhibitor and a much wider range of concentrations are needed. Errors needed on values. Kinetic study needs to quote errors on the Ki values and it looks like the values are reported to a much higher level of precision than is warranted (these types of experiments generally have errors of 5 to 10%). Experiments use mushroom enzyme which is known to have very different properties to the human enzyme (but to be fair many studies use the mushroom enzyme). The results are not very different from similar studies (which show the same type of inhibition and potency of inhibition). Ki values of 100 µM are poor potency and will be pharmacologically useless (especially given that many inhibitors of mushroom enzyme are inactive against the human enzyme, even assuming there are no issues with uptake into cells etc.). It is well known that the Lineweaver-Burk plot is the worst possible method for analysing enzyme kinetic data and the authors should provide in addition the following graphs: Direct (Michaelis-Menten) plot; Direct Linear Plot; Hanes plot; Eadie-Hofstee plot; Residuals plot of rates vs. substrate and inhibitor concentrations. The authors assert this is a reversible inhibitor but provide no evidence to support their claim e.g. a rapid dilution experiment showing restoration of activity following incubation of concentrated enzyme with concentrated inhibitor.
What is the point of the fluorescence quenching experiments? There is a lot of detail but it doesn’t tell the reader what conclusions are reached.
Experimental
No sources of materials or general experimental is given. The synthesis is a trivial one-step reaction. Insufficient details are given to repeat the experiment (for example, what TLC conditions were used?).
Enzyme assay needs to specify what type of sodium phosphate buffer was used (there are three different ways it can be made) and what concentration. Details of the computation methods need to be given (what programme and version? How many repeats for data? What substrate and inhibitor concentrations were used? How were rates derived from the data?).
Other issues
Albinism is caused by a reduction / absence of melanin not the over-production.
Badly written opening paragraph. Mixing tenses and jumping between subjects.
Reviewer 2 Report
In this work, Wang and coworkers reported the preparation of 2-(2,6-dimethylhepta-1,5-dien-1-yl)quinazolin-4(3H)-one as a novel tyrosinase inhibitor. The synthesis relies on the condensation between citral and o-amino-benzamide. I have some suggestions:
-the reaction was performed in DMSO. I would like to ask if the authors tried to perform this reaction using greener solvents;
-Q1 has been purified by column chromatography. What about the retention factor (Rf) of the molecule?
-in section 3.1 the authors stated: "...using CDCl3-d6 as an internal standard". What does it mean?
-the IR analysis of the molecule is missing;
-in section 2.1 "yeild" should be changed with "yield";
- Looking at chart1, the purity of compound Q1 seems to be satisfactory. However, according to me, the 1H and 13C spectra should be better aligned in the chart1.
I can't give any suggestions regarding the biological evaluation of Q1.